# Sick of news? Television news exposure, collective stressful events and headache related emergency department visits

**Alina Vodonos[1], Victor Novack[1], Israel Waismel-Manor**  **[2]\*, Yacov Ezra[3], Adi Guetta[3], Gal Ifergane[3]**

**1** Clinical Research Center, Soroka University Medical Center, Beer-Sheva, Israel, **2** School of Political Science, University of Haifa, Haifa, Israel, **3** Neurology Department, Soroka University Medical Center, Beer-Sheva, Israel

\* wisrael@poli.haifa.ac.il

**Data Availability Statement:** Data cannot be shared publicly because of medical privacy restrictions. Data are available from the Soroka clinical research center to the reviewers and for all

## Abstract

Stress is a well-known trigger for primary headache yet its impact is difficult to demonstrate in large epidemiological studies. Israeli national TV news is often referred to as the "tribal fire", as many Israelis watch national news coverage following terror attacks or military operations. We examined the association between exposure to television news and their content with headache related Emergency Department visits. This retrospective cohort study included data on daily Emergency Department visits with a chief complaint of headache in Soroka University Medical Center, during 2002–2012. Data on daily television news viewership ratings were obtained from the Israeli Audience Research Board and its content from Channel 2 headlines, the highest rated TV news program. To estimate the short-term effects of news rating during the evening news on the number of daily headache visits, we applied generalized linear mixed models. 16,693 Emergency Department visits were included in the analysis. An increase in five units of daily rating percentages was associated with increase in Emergency Department visits the following day, relative risk (RR) = 1.032, (95% CI 1.014–1.050). This association increased with the age of the patients; RR = 1.119, (95% CI 1.075–1.65) for older than 60-year-old, RR = 1.044 (95% CI 1.010–1.078) for ages 40–60 and RR = 1.000 (95% CI 0.977–1.023) for younger than 40-year-old. We did not find a specific content associated with ED visit for headache. Higher television news ratings were associated with increased incidence of Emergency Department headache related visits. We assume that especially among older persons, news viewership ratings provide an indirect estimate of collective stress, which acts as a headache trigger for susceptible subjects.

## Introduction

Non-traumatic headache episodes are a common presenting complaint at emergency departments worldwide [1]. Intracranial pathology is found in only a minority of the patients, and it is therefore believed that most of the ED visits represent primary headache disorders [2].

researchers who meet the criteria for access to confidential data. Data requests may be sent to Michal Gordon, Ph.D. Director of Data Research at the Soroka Clinical Research Center: Michalgord@clalit.org.il.

**Funding:** The authors received no specific funding for this work.

**Competing interests:** The authors have declared that no competing interests exist.

Many intrinsic precipitating factors for headache episodes in susceptible individuals were reported [3–8]: afferent stimulation, physical activity, menstrual changes, psychological factors, sleep cycle disorders, and dietary factors, yet none of them can fully account for the variability in the frequency of ED headache related visits. Extrinsic factors such as climate patterns and ambient air pollution was previously reported to be associated with changes in daily headache related ED visits as well [9].

Patients, with both tension and migraine type headaches, often report stress to be one of their major triggers [4, 10]. Although stress is a universal experience, it is difficult to measure the personal level of stress, and it is almost impossible to quantify the level of collective stress, experienced by groups or communities.

A potential source of stress may be the news, particularly bad news. According to Affective Intelligence—AI [11], anxious individuals are motivated to be information seekers, as information may be vital to their own survival. Hence, it is not surprising, that while individuals state they do not appreciate negative news, we witness a tendency to seek such news [12].

News watching in itself might elevate anxiety. A study from the USA where self-reported data was collected, suggested that following the news was one of the biggest daily stressors among individuals who report a high level of stress [13]. Moreover, a recent internet-based survey following the Boston Marathon bombing showed that repeated bombing-related media exposure was associated with higher acute stress than was direct exposure (presence in the bombing area). In addition, watching a 15-minute news segment was enough to trigger persisting negative psychological feelings, which decreased only after a progressive relaxation [14].

In this ecological population based study we sought to evaluate the association between the daily national news consumption and their content with headache related ED visits.

## Materials and methods

### Study population

The Soroka University Medical Center (SUMC) internal review board approved the study—0361–12. Due to the nature of this study, patients did not sign a consent form. We correlate ER visits to TV ratings using no personal patient information. We obtained data on daily ED visits to SUMC with a primary complaint of headache in patients 18 years or older during a ten-year period (between December 1, 2002 and December 31, 2011). SUMC is a tertiary 1000 bed hospital and the only medical center in the Negev region, serving a population of 700,000 residents. The annual adult ED visits volume is ~100,000. We excluded from the analysis patients hospitalized following the ED visit as we presumed them to have a serious medical problem requiring ED visit unrelated to the stress trigger.

### Television news rating data

National Israeli Television news exposure levels were obtained from the Israeli Audience Research Board (IARB), which measures TV viewership ratings using a "people meter" method. In this method, electronic tracking devices are connected to all the television sets in the homes of the panel participants, currently numbering 580 households with approximately 1800 individuals, comprising a representative sample of the television owners in Israel. The device monitors every act of turning on, turning off, and switching between channels on the television sets. Monitoring is performed in real time and continuously throughout the day, 24/7. Daily viewing ratings of the evening news edition in the three major TV news channels were obtained for the study period. Overall, television news consumption in Israel is quite high: 37.3% of Israelis watched in 2012 the evening news [15], in comparison to 27% of Americans who watched the networks' evening news in the USA [16].

## Television news content analysis

To explore which news items generated the highest rating, we collected the first two headlines of Channel 2 evening news, the most watched evening news program, with 60% of all television nightly news viewership [15]. Headline data was available from 2008 till 2012. Two trained coders rated all the first two opening headlines from Channel 2 evening news. Headlines were categorized into seven non-exclusive categories. Categorization of 2,421 headlines into seven non-exclusive categories was performed by two coders which obtained a percent agreement of 92.3%, Cohen's κ = 0.578 (95% CI, 0.266 to 0.830), p < 0.0001. These categories are Military Conflict and Terrorism (military operations by or against Israel and terrorist attacks against Israelis), Israeli Politics (legislation, campaigns, etc.), Israeli Foreign Relations (peace negotiations, high level meetings or exchanges with foreign leaders and diplomats), Domestic Crime and Scandals, the Economy, World Events (disasters, foreign elections, wars, etc.) and Miscellaneous (all items which did not fit all other categories.

## Statistical analysis

Data were summarized using frequency tables for categorical variables and summary statistics (median with inter-quartile range, mean with standard deviation and range) for continuous variables. To estimate the short term effects of daily news rating on number of daily headache ED visits, we applied a generalized linear mixed models (GLMM) methodology described by Szyszkowicz M. et. al. [17]. Poisson random-effects models were utilized to analyze the cluster counts where the groups of days, determined by the triplet (day of the week, month, year) formed the clusters. The hierarchical construction of the clusters allows the model to incorporate level specific random effects. We used the glmmPQL function from the R statistical package to perform the analysis. Relative risk (RR) with 95% confidence intervals (CI) represent the risk for of ED visits for headache associated with an increase in daily news rating adjusted for public holiday days. Additionally, we stratified our analysis by gender and by age groups (<40, 40–60 and >60 years old) to compare the magnitude of the effect. Scatter plot with overlay of LOESS (locally weighted scatter plot smoothing) curve was produced to examine the relationship between daily estimation and predicted daily ED visits with headache. P values less than 0.05 were considered statistically significant. We used R statistical package, v 3.2 to perform the analysis.

## Results

The study population comprised 14,010 patients with 16,693 ED visits, with 19.1% of the patients having more than one visit during the study decade. Among them, 57.6% female and 54.3% were younger than 40 years old (Table 1). Mean, median, minimum and maximum daily television rating percentages (the percent of potential viewers who watched the evening

**Table 1. Frequency of ED visits for headache (2002–2012).**

| | |
|---|---|
| **Total Number of Headache ED visits** | 16,693 |
| *Age (years), mean ± SD* | 41.3 ±18.6 |
| **<40, n (%)** | 9,067 (54.3%) |
| **40–60, n (%)** | 4,608 (27.6%) |
| **>60, n (%)** | 3,018 (18.1%) |
| *Gender* | |
| **Female, n (%)** | 9,616 (57.6%) |
| **Number of visits per patients, median (Range)** | 1 (1–31) |

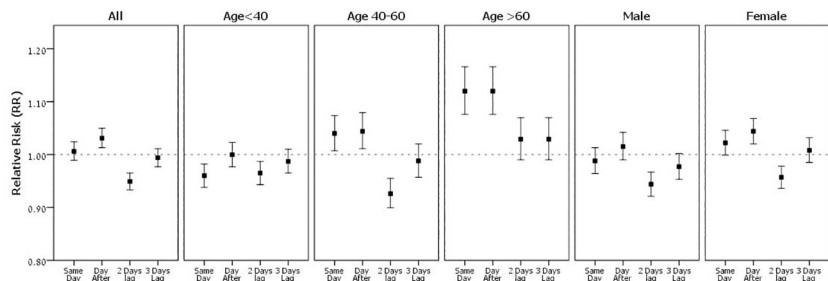

**Fig 1. Relative Risk (RR) for daily ED visits for headache associated with daily rating percent, among age groups and by male and females.** Relative Risk (RR) and 95% Confidence Interval for ED visits per increase in 5 units of daily rating percentages. Results of the separated Poisson regression models, for study period 2002–2012, adjusted for public holidays.

news) of all three major news channels during the study period were the following; 29.3±5.2, 29.5, 3 and 51.0, respectively.

## Effect of daily news rating on the incidence of headache ED visits

An increase of 5% in daily rating was associated with increase in ED visits the following day, relative risk (RR) = 1.017, (95% CI 1.001–1.033). (Fig 1) but not two and three days after, RR = 0.953(95% CI 0.939–0.967) and RR = 0.990 (95% CI 0.973–1.007), respectively, where the association was protective. This association increased with the age of the patients; RR = 1.119, (95% CI 1.075–1.65) for older than 60-year-old, RR = 1.044 (95% CI 1.010–1.078) for ages 40–60 and RR = 1.000 (95% CI 0.977–1.023) for younger than 40-year-old. In addition, we observed a marginal increase in the effect among female patients, RR = 1.04 (95% CI 1.020–1.068) as compared to men, RR = 1.015 (95% CI 0.990–1.042). The non-linear association between rating estimation the following day and ED visits with headache was expressed as LOESS curve scatter plot (S1 Fig).

Similar to the linear analysis, the magnitude of effects observed was larger among older patients. We further examined whether two days of cumulative exposure to high percentage of rating estimates (above 29.5%—median rating percentage) is associated with an additional increase in risk of ED admission. This effect was found only among older patients (over the age of 60); RR = 1.275 (1.172–1.387) (Table 2).

## Content analysis of leading evening news items

News broadcasts that open with reports on military operations and terror attacks against civilians were the most watched news editions over this period (top 10% rating headlines) (S2 Fig). We further analysis the magnitude of the effect of increase in ED visits the following day across

**Table 2. Relative Risk (RR) for daily ED visits for headache associated with cumulative exposure over 29.5%* percent in two days preceding the ED visit.**

|  | RR** | 95%CI |
|---|---|---|
| **All** | 0.990 | 0.952–1.030 |
| **Age >60** | 1.275 | 1.172–1.387 |
| **Age 40–60** | 1.039 | 0.967–1.116 |
| **Age <40** | 0.884 | 0.839–0.933 |
| **Male** | 0.958 | 0.905–1.015 |
| **Female** | 1.019 | 0.968–1.072 |

different content of the headlines (S1 Table). We did not find a specific content associated with ED visit for headache.

## Discussion

In this analysis of more than fifteen thousand headache related ED visits over a decade, we attempted to assess the short term effects of Israeli national news consumption. We demonstrated an association between news consumption and the incidence of headache related ED visits (during the following day. This association was particularly pronounced in patients over 60 years of age. This effect was not related to the content of the news headlines.

Stress is a well-recognized headache trigger, in both migraine and tension type headache. Recently, in a longitudinal study [18] self-report of stress was associated with the frequency of both migraine and tension type headache (an increase of 10 points in stress intensity measured by VAS 0–100 was associated with an increase of 6.0% of TTH frequency and 4.3% for migraine). The effect was higher in younger age groups. Increased headache prevalence was also reported among survivors and responders following major natural or man-made disasters [19, 20].

While stress is a subjective experience affected by multiple biological, psychological and social factors, collective community wide experiences may affect the personal level of stress. The effect of such collective experiences is usually referred to as collective stress [21].

During major national or international crises, the public seeks information. This public attention, which elevates news consumption, may be indicative of increased stress levels. In this sense, the level of news exposure, gauged by television news ratings, can be perceived as a surrogate marker of increased collective stress. On the other hand, widespread media coverage transmits the impact of stressful events to populations not exposed to the event itself, amplifying collective stress and psychological trauma related symptoms. Such effect was demonstrated in several major events and disasters such as the Oklahoma City bombing [22], H1N1 influenza pandemic [23], 1994 California earthquake [24], Boston Marathon bombing [14], the 1990 Gulf War [25], and 9/11 [26]. Therefore, news consumption is not only a marker of collective stress, it is also a collective stress amplifier, contributing by itself to the stress and its effect.

Two mechanisms may explain the association between the collective stress and increased rates of headache associated ED visits. The first is a psychobiological effect, according to which, collective stress is a headache-precipitating factor similar to personal stress, possibly increasing the likelihood of headaches or their severity, resulting in increased rates of ED consultations. The second is a psychosocial effect, increasing the tendency of headache sufferers and their caregivers to seek emergency medical support for headache episodes during high collective stress events.

While the association between personal stress and different types of headaches was especially pronounced in young individuals, in our study, the association was the greatest among individuals older than 60 and had a marginal increase in the effect among female patients. Since TV news exposure is largely influenced by age, with higher exposure rates in the elderly population [27], it is possible that news viewing rates best reflect (or amplify) collective stress levels among older age groups, and not among younger individuals which mostly consume news from new media rather than television. It is also possible that elderly individuals with smaller support networks have a higher tendency to seek emergency medical assistance in high collective stress situations. Moreover, it was suggested that women suffer more stress than men and their coping style is more emotion-focused compared to men [28].

Our study has several important limitations. First, our study does not allow conclusions about the interaction between media consumption, collective stress and headaches. Moreover, Israel is unique in its continuing stressful events lifestyle and its media consumption style, therefore making the generalization of the results to other countries and communities difficult [29]. The nature of the data does not allow differentiating potential differences between TTH and migraine. Lastly, the ecological nature of our study does not allow for the individual assessment of news exposure.

The association of media consumption and headache morbidity should be further studied in different communities and setups. Prospective assessment of this association may provide us with new insights on the association between headache suffering and collective stress and possibly provide us with another possibly modifiable headache trigger.

## Supporting information

**S1 Fig. Predicted daily ED visits with headache associated with daily rating percent, among age groups and by male and females.** Relative Risk (RR) and 95% Confidence Interval for ED visits per increase in 5 units of daily rating percentages. Results of the separated Poisson regression models, for study period 2002–2012, adjusted for public holidays.
(TIF)

**S2 Fig. News headline categories by television ratings.**
(TIF)

**S1 Table. Relative Risk (RR) for daily ED visits for headache associated with daily raiting percent, by different content of the headlines.**
(DOCX)

## Author Contributions

**Conceptualization:** Alina Vodonos, Victor Novack, Israel Waismel-Manor, Yacov Ezra, Adi Guetta, Gal Ifergane.

**Data curation:** Alina Vodonos, Israel Waismel-Manor.

**Formal analysis:** Alina Vodonos.

**Investigation:** Israel Waismel-Manor.

**Methodology:** Alina Vodonos, Israel Waismel-Manor, Adi Guetta.

**Project administration:** Victor Novack, Gal Ifergane.

**Supervision:** Victor Novack, Gal Ifergane.

**Writing – original draft:** Alina Vodonos.

**Writing – review & editing:** Alina Vodonos, Israel Waismel-Manor.

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
