## [Decision Letter · Decision Letter 0]

10 Jul 2020

PONE-D-20-00517

Sick of news? Television news exposure and headache related emergency department visits

PLOS ONE

Dear Dr. waismel-manor,

Thank you for submitting your manuscript to PLOS ONE. After careful consideration, we feel that it has merit but does not fully meet PLOS ONE’s publication criteria as it currently stands. Therefore, we invite you to submit a revised version of the manuscript that addresses the points raised during the review process.

The manuscript has been evaluated by three reviewers, and their comments are available below.

The reviewers have raised a number of concerns that need attention. The reviewers have suggested changes to the abstract and further details regarding the primary outcome measure. As reviewer #1 notes, the study cannot disentangle the effects of watching the news and stressful events occurring within the community. Since either could explain the results, the title should be revised and this limitation addressed in the Introduction and Discussion sections. Regarding the title, my recommendation would be to remove “Sick of news?”.

Could you please revise the manuscript to carefully address the concerns raised?

We look forward to receiving your revised manuscript.

Kind regards,

George Vousden

Senior Editor

PLOS ONE

Journal Requirements:

Reviewers' comments:

Reviewer's Responses to Questions

**Comments to the Author**

1. Is the manuscript technically sound, and do the data support the conclusions?

Reviewer #1: Partly

Reviewer #2: Yes

Reviewer #3: Yes

2. Has the statistical analysis been performed appropriately and rigorously? 

Reviewer #1: Yes

Reviewer #2: Yes

Reviewer #3: Yes

3. Have the authors made all data underlying the findings in their manuscript fully available?

Reviewer #1: No

Reviewer #2: Yes

Reviewer #3: Yes

4. Is the manuscript presented in an intelligible fashion and written in standard English?

Reviewer #1: Yes

Reviewer #2: Yes

Reviewer #3: Yes

5. Review Comments to the Author

Reviewer #1: This manuscript presents the ecologic association of television news ratings and emergency department visits for headache in a tertiary care hospital in Israel. Headache visits were found to be slightly more common on the day following a day with higher television news ratings. The association was slightly stronger among older adults.

Television news ratings seem to be conceptualized as both a marker of community stressful events and as a potential cause of stress. It would be helpful to state this more clearly in the introduction. Given the inability to differentiate between the two possible mechanisms, I would revise the title.

Please provide more detail on how primary complaint of headache was identified. Is this based on a standardized coding system?

It would be helpful to formally test for interaction of news ratings with age and gender.

It is not clear to me what exactly is presented in Supplemental Figures 1 and 2.

I did not have access to the supplemental table to review.

Line 219. Please elaborate on the unique aspects of media consumption in Israel.

Reviewer #2: This study evaluated the association between news category and ER headache visit. The authors found that an increase in five units of daily rating percentages was associated with increase in Emergency Department visits the following day. I am hoping to comment several points.

1. The authors described that they categorized headline news by category. However, there was only few descriptions for "increase in 5 units of daily rating percentage". Consider revision for more description for this issue at METHODS for better under standing

2. The authors analyzed the association between all headaches and news contents. Two common primary headaches, migraine and tension-type headache, have different triggering factors in addition to clinical differences. Therefore, migraine and TTH may show different pattern to stressful news. It would be more valuable if additional analyses for the difference in the association between ER headache visit and stressful news. If it is not possible, consider additional description as a limitation of this study.

3. One of limitation of this study is the setting of the study. The data of this study were collected in a single university hospital in Israel. Therefore, it is difficult to generalize. This could be a limitation of this study.

4. Trivia. Decimal point issue. Some decimal points were needed to revise collectively. 41.25±18.6 -> 41.3±18.6.

Reviewer #3: The authors examined the relation between TV news and emergency department visits because of headache.

The study covered almost a decade and included a total of 16,693 emergency department visits.

The study is interesting, but study objectives and methods require more clarity and the authors should avoid selective reporting in the abstract.

In the abstract and in the introduction, the authors should specify the objectives of the study, namely to analyze the relation between emergency room visits because of headache and (1) the daily rate of persons watching TV news and (2) the content of the news.

Throughout the paper, the authors should avoid phrases such as “rating percentages”. Such terms are ambiguous and leave open what the rating refers to.

The lack of an association between news content and emergency room visits because of headache must be reported in the abstract.

The low rate of TV news consumers (less then 30 %) should also be reported in the abstract and mentioned in the discussion.

In the abstract and in the results, the authors should check following finding, “This association increased with the age of the patients; RR =1.119, (95% CI 1.014; 1.050) …“ The confidence interval cannot be smaller than the RR.

In the paragraph on the limitations of the study, the authors should replace the general statement, “Our ability to understand the interaction between media consumption, collective stress and headache in relation to the different news contents is limited” by a statement specifically related to their study such as, “Our study does not allow conclusions about the interaction between media consumption, collective stress and headache.”

The English text needs some editing.

6. PLOS authors have the option to publish the peer review history of their article (what does this mean?). If published, this will include your full peer review and any attached files.

Reviewer #1: No

Reviewer #2: No

Reviewer #3: **Yes: **Prof Çiçek Wöber-Bingöl, MD

---

## [Author Response · Author response to Decision Letter 0]

30 Oct 2020

September 19th, 2020

Manuscript #: PONE-D-20-00517

Entitled; Sick of news? Television news exposure and headache related emergency department visits

Dear Dr. Joerg Heber, PhD

Editor in chief, PLOS ONE

Please see our revised submission of the manuscript entitled "Sick of news? Television news exposure and headache related emergency department visits." We have carefully reviewed the comments of the reviewers, and would like to thank them for their thoughtful suggestions, which we think have substantially improved the quality of our manuscript. We have addressed the comments below in this letter, and in the revised manuscript attached. Please note, we have submitted both the revised version with the ‘track changes’ mode, as well as a ‘clean’ copy.

We sincerely hope the revisions made will meet your approval. 

Looking forward to hearing from you,

Israel Waismel-Manor, PhD

Corresponding author

Reviewer #1: This manuscript presents the ecologic association of television news ratings and emergency department visits for headache in a tertiary care hospital in Israel. Headache visits were found to be slightly more common on the day following a day with higher television news ratings. The association was slightly stronger among older adults.

Comment:

Television news ratings seem to be conceptualized as both a marker of community stressful events and as a potential cause of stress. It would be helpful to state this more clearly in the introduction. Given the inability to differentiate between the two possible mechanisms, I would revise the title.

Answer:

Thank you for this. Indeed, it is a correlation and the previous title implies there is one. Therefore, we changed it to “Sick of news? Television news exposure, collective stressful events and headache related emergency department visits”.

We used the phrase “Sick of News” as a catchy phrase, a trigger, but the question mark following it indicates we have no causal claim, and the revised second part of the title is now even clearer. 

Comment:

Please provide more detail on how primary complaint of headache was identified. Is this based on a standardized coding system?

Answer:

We analyzed admitted to ED (emergency department) with a chief complaint of headache, but excluded from the analysis all cases that were hospitalized following the ED visit.

Comment:

It would be helpful to formally test for interaction of news ratings with age and gender.

Answer:

We thank the reviewer for this suggestion. However, since this is an ecological /aggregated time series analysis, (not individual age and gender units) we are limited in our ability to test the interaction of individual characteristics beyond the method of examining the magnitude of the effect within a subset of different age and gender groups. 

Comment:

It is not clear to me what exactly is presented in Supplemental Figures 1 and 2.

I did not have access to the supplemental table to review.

Answer:

Supplemental Figure 1 presents the Relative Risk (RR) and 95% Confidence Interval for ED visits per increase in 5 units of daily rating percentages for all, different age groups and per different genders. Results of the separated Poisson regression models, for study period 2002-2012, adjusted for public holidays 

Supplemental Figure 2- presents the “News Headline Categories by Television Ratings” Subsection “Content Analysis of Leading Evening News Items” was rewritten.

Comment:

Line 219. Please elaborate on the unique aspects of media consumption in Israel.

Answer:

We appreciate pointing it out. Sometimes we are so immersed in our own case that we forget it is meant for a general audience. We have modified the “Television News Rating Data” and the “Television News Content Analysis” to explain that 1) the 8 pm evening TV news consumption is Israel is quite high (almost 40% more than the and 6:30 pm evening news in the US, and that 2) Channel 2, the Channel selected for the content analysis is the most watched (and best represents) of the three channels, controlling 60% of the market. 

Comment:

Reviewer #2: This study evaluated the association between news category and ER headache visit. The authors found that an increase in five units of daily rating percentages was associated with increase in Emergency Department visits the following day. I am hoping to comment several points.

1. The authors described that they categorized headline news by category. However, there was only few descriptions for "increase in 5 units of daily rating percentage". Consider revision for more description for this issue at METHODS for better under standing

Answer: Association between increase in 5 units of daily rating percentage per different content of the headline groups was presented in Table 1 in the supplemental material. 

Comment:

2. The authors analyzed the association between all headaches and news contents. Two common primary headaches, migraine and tension-type headache, have different triggering factors in addition to clinical differences. Therefore, migraine and TTH may show different pattern to stressful news. It would be more valuable if additional analyses for the difference in the association between ER headache visit and stressful news. If it is not possible, consider additional description as a limitation of this study.

Answer:

As stated in the introduction and methods sections, headache phenotype is not available for analysis in this analysis. We elaborated on the effects of stress on both types of primary headaches. According to the reviewers’ suggestion, we listed it as a limitation of the study in the discussion section. 

Comment:

3. One of limitation of this study is the setting of the study. The data of this study were collected in a single university hospital in Israel. Therefore, it is difficult to generalize. This could be a limitation of this study.

Answer:

As we mentioned in the text, we obtained data on daily ED visits to Soroka University Medical Center (SUMC). SUMC is a tertiary 1000 bed hospital and the only medical center in the Negev region, and provides medical services to the over one million residents of the region which covers 60% of the geographical area of the country. Thus, this unique setting resolves the generalization question.

Comment:

4. Trivia. Decimal point issue. Some decimal points were needed to revise collectively. 41.25±18.6 -> 41.3±18.6.

Answer:

Thank you for this point. We revised the manuscript accordingly.

Reviewer #3: The authors examined the relation between TV news and emergency department visits because of headache. The study covered almost a decade and included a total of 16,693 emergency department visits.

The study is interesting, but study objectives and methods require more clarity and the authors should avoid selective reporting in the abstract.

Comment

In the abstract and in the introduction, the authors should specify the objectives of the study, namely to analyze the relation between emergency room visits because of headache and (1) the daily rate of persons watching TV news and (2) the content of the news.

Answer:

We thank the reviewer for suggesting this clarification of the objectives of the study. We rephrased the abstract the introduction in the following phrasing: 

“In this ecological population based study we sought to evaluate the association between the daily national news consumption and their content with headache related ED visits. “

Throughout the paper, the authors should avoid phrases such as “rating percentages”. Such terms are ambiguous and leave open what the rating refers to.

The lack of an association between news content and emergency room visits because of headache must be reported in the abstract.

Answer:

We added the finding of lack of an association between news content and emergency room visits in the abstract as well as in the full text. 

Comment

The low rate of TV news consumers (less than 30 %) should also be reported in the abstract and mentioned in the discussion.

Answer:

We assume this 30% estimate derives from the explanation about the people meter, which monitors 580 representative Israeli households, with an average of about 3 residents per household, with a total of about 1800 individuals in them (TV News ratings for this period were about 37% of the population). This rating is actually quite high in comparison to other countries like the US, where ratings were about 27% at that time. 

Even today, after a major shift to news online and on social media and other mobile platforms, evening TV news viewership in Israel is about 15% of the Israeli population (https://www.the7eye.org.il/rating/387893 ). In the US the number of Americans who are currently watching the evening news in all major networks combined is 31 million, or less than 10% (https://www.poynter.org/newsletters/2020/america-is-watching-the-evening-news-again-tv-news-numbers-are-up-way-up/ ) and in Britain, BBC news, the most viewed TV news program, has an average audience of 5.2 million viewers, less than 8% of its population, (https://pressgazette.co.uk/coronavirus-leads-to-staggering-demand-for-trusted-tv-news/). 

We analyzed Channel 2 headlines because it is the most watched news program on Israeli television, with an average viewership at that time of about 22.4% of the total population (with Channel 10 holding 10.4% and Channel 1 with 4.9%). With such a market domination, we are confident that Channel two still serves as Israel’s “tribal fire” and is quite indicative of the national media menu Israelis consume, representing the topics that capture the public’s interest on a given day. We have clarified the abstract and the body of the manuscript to explain this.

Comment

In the abstract and in the results, the authors should check following finding, “This association increased with the age of the patients; RR =1.119, (95% CI 1.014; 1.050) …“ The confidence interval cannot be smaller than the RR.

Answer:

Thank you for pointing out, this is indeed a typo. The true confidence interval is RR =1.119, (95% CI 1.075; 1.65) . We revised the manuscript accordingly. Page 2 and Page7.

Comment

In the paragraph on the limitations of the study, the authors should replace the general statement, “Our ability to understand the interaction between media consumption, collective stress and headache in relation to the different news contents is limited” by a statement specifically related to their study such as, “Our study does not allow conclusions about the interaction between media consumption, collective stress and headache.”

Answer:

This is an important point. While we intended to suggest that this is a correlative study, and that we cannot make causal inferences, this sentence was not clear enough. We adopted the reviewer’s wording to better reflect our study’s limitations.

---

## [Decision Letter · Decision Letter 1]

4 Feb 2021

PONE-D-20-00517R1

Sick of news? Television news exposure, collective stressful events and headache related emergency department visits

PLOS ONE

Dear Dr. waismel-manor,

Thank you for submitting your manuscript to PLOS ONE. After careful consideration, we feel that it has merit but does not fully meet PLOS ONE’s publication criteria as it currently stands. Therefore, we invite you to submit a revised version of the manuscript that addresses the points raised during the review process.

We look forward to receiving your revised manuscript.

Kind regards,

Junaid Ahmad Bhatti

Academic Editor

PLOS ONE

Additional Editor Comments (if provided):

The revised version has received favorable reviews. Kindly revise the manuscript as per the suggestions for minor revisions.

Reviewers' comments:

Reviewer's Responses to Questions

**Comments to the Author**

1. If the authors have adequately addressed your comments raised in a previous round of review and you feel that this manuscript is now acceptable for publication, you may indicate that here to bypass the “Comments to the Author” section, enter your conflict of interest statement in the “Confidential to Editor” section, and submit your "Accept" recommendation.

Reviewer #2: All comments have been addressed

Reviewer #3: All comments have been addressed

2. Is the manuscript technically sound, and do the data support the conclusions?

Reviewer #2: Yes

Reviewer #3: Yes

3. Has the statistical analysis been performed appropriately and rigorously? 

Reviewer #2: Yes

Reviewer #3: Yes

4. Have the authors made all data underlying the findings in their manuscript fully available?

Reviewer #2: Yes

Reviewer #3: No

5. Is the manuscript presented in an intelligible fashion and written in standard English?

Reviewer #2: Yes

Reviewer #3: Yes

6. Review Comments to the Author

Reviewer #2: The authors properly addressed all the points mentioned in the previous review. The authors also adequately revised the manuscript after the previous review.

Reviewer #3: The authors followed the reviewer's suggestions. Only one sentence in the abstract and results still needs revision.

"This association increased with the age of the patients; RR =1.119, (95% CI 1.075; 1.065) for older than 60-year-old, RR=1.044 (95% CI 1.010-1.078) for ages 40-60 and RR= 1.000 (95% CI 0.977-1.023) for younger than 40-year-old.

There is still a problem with the 95% CI in the older persons!

7. PLOS authors have the option to publish the peer review history of their article (what does this mean?). If published, this will include your full peer review and any attached files.

Reviewer #2: No

Reviewer #3: **Yes: **Prof Çiçek Wöber-Bingöl, MD

---

## [Author Response · Author response to Decision Letter 1]

1 Mar 2021

Reviewer #3: The authors followed the reviewer's suggestions. Only one sentence in the abstract and results still needs revision.

"This association increased with the age of the patients; RR =1.119, (95% CI 1.075; 1.065) for older than 60-year-old, RR=1.044 (95% CI 1.010-1.078) for ages 40-60 and RR= 1.000 (95% CI 0.977-1.023) for younger than 40-year-old.

There is still a problem with the 95% CI in the older persons!

Response:

We kindly apologize for not fixing the typo: The true confidence interval for older persons is RR =1.119, (95% CI 1.075; 1.65) . We revised the manuscript accordingly. Page 2 and Page 7.

---

## [Editor Report · Decision Letter 2]

25 Mar 2021

Sick of news? Television news exposure, collective stressful events and headache related emergency department visits

PONE-D-20-00517R2

Dear Dr. waismel-manor,

We’re pleased to inform you that your manuscript has been judged scientifically suitable for publication and will be formally accepted for publication once it meets all outstanding technical requirements.

Kind regards,

Junaid Ahmad Bhatti

Academic Editor

PLOS ONE
---

## [Editor Report · Acceptance letter]

29 Mar 2021

PONE-D-20-00517R2 

Sick of news? Television news exposure, collective stressful events and headache related emergency department visits 

Dear Dr. Waismel-Manor:

I'm pleased to inform you that your manuscript has been deemed suitable for publication in PLOS ONE. Congratulations! Your manuscript is now with our production department. 

Kind regards, 

on behalf of

Dr. Junaid Ahmad Bhatti 

Academic Editor

PLOS ONE